# Dysregulated Ca^2+^ Homeostasis as a Central Theme in Neurodegeneration: Lessons from Alzheimer’s Disease and Wolfram Syndrome

**DOI:** 10.3390/cells11121963

**Published:** 2022-06-18

**Authors:** Manon Callens, Jens Loncke, Geert Bultynck

**Affiliations:** KU Leuven, Laboratory of Molecular & Cellular Signaling, Department of Cellular & Molecular Medicine, Campus Gasthuisberg, O/N-I bus 802, Herestraat 49, BE-3000 Leuven, Belgium; manon.callens@kuleuven.be (M.C.); jens.loncke@kuleuven.be (J.L.)

**Keywords:** calcium, mitochondria, mitochondria-associated ER membranes (MAMs), Alzheimer’s disease, Wolfram syndrome, neurodegeneration

## Abstract

Calcium ions (Ca^2+^) operate as important messengers in the cell, indispensable for signaling the underlying numerous cellular processes in all of the cell types in the human body. In neurons, Ca^2+^ signaling is crucial for regulating synaptic transmission and for the processes of learning and memory formation. Hence, the dysregulation of intracellular Ca^2+^ homeostasis results in a broad range of disorders, including cancer and neurodegeneration. A major source for intracellular Ca^2+^ is the endoplasmic reticulum (ER), which has close contacts with other organelles, including mitochondria. In this review, we focus on the emerging role of Ca^2+^ signaling at the ER–mitochondrial interface in two different neurodegenerative diseases, namely Alzheimer’s disease and Wolfram syndrome. Both of these diseases share some common hallmarks in the early stages, including alterations in the ER and mitochondrial Ca^2+^ handling, mitochondrial dysfunction and increased Reactive oxygen species (ROS) production. This indicates that similar mechanisms may underly these two disease pathologies and suggests that both research topics might benefit from complementary research.

## 1. Introduction

Neurodegenerative diseases are characterized by the progressive degeneration or loss of neurons, a process that is causative of a wide array of symptoms. In this review, we discuss the important role of Ca^2+^ homeostasis in two neurodegenerative diseases: Alzheimer’s disease and Wolfram syndrome. The main focus lies on the ER and mitochondrial Ca^2+^ handling, as well as the close interaction between these organelles. By investigating the Ca^2+^ impairments, we show that these two very different diseases have more in common than first meets the eye.

## 2. Ca^2+^ Signaling

### 2.1. Ca^2+^ Signaling: A Brief Overview

Calcium ions (Ca^2+^) are omnipresent in living cells and function as a versatile form of cellular communication, vital for the regulation of numerous cellular processes, including muscle contraction, fertilization, differentiation and metabolism [1,2]. For instance, several metabolic enzymes function with Ca^2+^ as a co-factor, including pyruvate-, isocitrate- and α-ketoglutarate dehydrogenase. In this way, a Ca^2+^ transfer from the endoplasmic reticulum (ER) to mitochondria can stimulate mitochondrial ATP production and survival [3]. A few examples of the involvement of Ca^2+^-dependent processes in specific tissue/cell types are shown in Figure 1B–D. The living cells maintain a relatively low cytosolic Ca^2+^ concentration ([Ca^2+^]) (100 nM), compared to the [Ca^2+^] inside the intracellular Ca^2+^-storing organelles (250–500 µM) and the extracellular space (>1 mM). The rapid changes in the cytosolic [Ca^2+^] can relay signals from either extracellular or intracellular stimuli to cellular effectors [4]. A critically controlled balance of the intracellular Ca^2+^ homeostasis is facilitated by the extensive ‘Ca^2+^ toolkit’ (Figure 1A). This toolkit comprises of several Ca^2+^ transporters and Ca^2+^-binding proteins, thereby enabling the intracellular compartmentalization of Ca^2+^ and spatiotemporal organization of Ca^2+^ dynamics [2]. In the neurons, Ca^2+^ signaling plays a key role in synaptic transmission and plasticity, and the processes of learning and memory formation [5].

The major intracellular Ca^2+^-storage organelle is the ER, although the Golgi apparatus, the nuclear envelope and the lysosomes also serve as Ca^2+^ stores and contribute to the localized Ca^2+^ signaling. Moreover, the mitochondria can function as additional Ca^2+^ buffers upon sequestering elevations in the cytosolic Ca^2+^ [2]. The Ca^2+^ is mobilized from the ER into the cytosol, but also to the mitochondria, mainly via two different release channels: the inositol 1,4,5-trisphosphate (IP_3_) receptor (IP_3_R), which is the most ubiquitously expressed ER-resident Ca^2+^ release channel, and the ryanodine receptor (RyR) [6,7,8]. Both of the receptors consist of three different isoforms displaying distinct tissue-specific distribution patterns [7]. The IP_3_Rs are opened upon binding of IP_3_, a second messenger produced from phosphatidyl inositol 4,5-bisphosphate (PIP2), through the action of phospholipase C isoform, in response to the activation of G-protein-coupled receptors and tyrosine-kinase receptors by hormones, neurotransmitters, growth factors and/or antibodies [1]. In addition to this, the release of Ca^2+^ from the ER is enhanced by Ca^2+^ itself, in a process called Ca^2+^-induced Ca^2+^ release (CICR), which allows for the amplification of the cytosolic Ca^2+^ signals. Importantly, high cytosolic [Ca^2+^] can also inhibit ER Ca^2+^ release, illustrating the bell-curve shaped Ca^2+^ dependence of ER Ca^2+^ release [7]. Besides these intracellular Ca^2+^-release channels, a plethora of ER-resident proteins, including presenilins (PSENs), have been implicated in functioning as Ca^2+^-leak channels thereby mediating a constitutive passive Ca^2+^ leak from the ER Ca^2+^ stores [9,10,11]. For an extensive review of the ER-Ca^2+^ leak channels, we refer to [10]. In addition to this, the channels controlled by ER Ca^2+^ levels, such as TMCO1, have been identified [12]. The Ca^2+^-permeable TMCO1 channels are activated by high ER Ca^2+^ levels, thereby preventing ER Ca^2+^ overload. Conversely, the sarco-endoplasmic reticulum ATPase (SERCA) pump is responsible for the active maintenance or refilling of the internal Ca^2+^ stores [13]. Decreases in [Ca^2+^] in the ER cause a dissociation of Ca^2+^ from the Ca^2+^ sensors’ stromal interaction molecule (STIM1/STIM2) [14]. This results in the dimerization and redistribution of STIM proteins and the unfolding of their STIM1 Orai-activating region (SOAR) domains. The latter interact with the hexameric Orai1/2/3 channels, which are Ca^2+^-selective channels located on the plasma membrane enabling store-operated Ca^2+^ entry (SOCE) [14]. Recently, the helical M4x peptide of Orai was put forward as the target of interaction with the STIM SOAR domains, in which certain leucine residues in the M4x peptide were deemed to be critical for the interaction with SOAR [15]. The interaction of STIM with the Orai channels is aided by the STIM polybasic C terminus that binds acidic phospholipids at the plasma membrane, maintaining close contact between the ER and the plasma membrane. Consequently, the Orai channels are opened, thereby mediating store-operated Ca^2+^ influx from the extracellular environment into the cytosol and driving the SERCA-mediated refilling of the ER. Of note, the IP_3_Rs functionally interact with the STIM/Orai complexes in such a way that the Ca^2+^ release is coupled to appropriate Ca^2+^ entry. The N-terminal STIM EF-hand motifs in the ER lumen are thought to inhibit IP_3_R-mediated Ca^2+^ release [16]. Vice versa, the IP_3_R-mediated decreases in the lumenal [Ca^2+^] drive the clustering of STIM2 proteins and consequently augment the SOCE [16,17].

Besides the store-operated Ca^2+^ channels, other channels in the plasma membrane are responsible for Ca^2+^ entry, including the voltage-gated Ca^2+^ channels (VOC), receptor-operated Ca^2+^ channels (ROC), transient receptor potential (TRP) channels and arachidonic acid-regulated Ca^2+^ (ARC) channels [4]. On the other hand, the Ca^2+^ extrusion to the extracellular environment is mediated by the plasma membrane Ca^2+^ ATPase (PMCA) and the plasma membrane Na^+^/Ca^2+^ exchanger (NCX). The PMCA displays a higher Ca^2+^ affinity but lower capacity for Ca^2+^ transport, while the NCX shows the opposite properties [5,18].

The final aspect of the ‘Ca^2+^ toolkit’ consists of the Ca^2+^-binding proteins that are crucial for the regulation of Ca^2+^ concentration in the cytosol and in the intracellular stores. The fine regulation of Ca^2+^ dynamics by the Ca^2+^-binding proteins is not only exerted by Ca^2+^ buffering, but also through direct interaction with the target proteins (such as the IP_3_R [19,20]) [21]. Examples of such Ca^2+^-binding proteins are calmodulin (CaM), which is able to bind four different Ca^2+^ ions, calbindin D-28 (CB-D28K), calretinin (CR) and parvalbumin (PV) [5,22]. Inside the ER, Ca^2+^ is buffered by calreticulin (CRT) and calnexin (CNX). More specifically for neurons, fourteen neuronal Ca^2+^ sensor proteins have been identified, such as hippocalcin, recoverin, neurocalcin-δ and GCAP1-3 [23]. These are important for the regulation of neuronal function and for the maintenance of neuronal health [22]. In addition, neuronal Ca^2+^-binding proteins have been described that also directly interact with the IP_3_R [24].

### 2.2. Mitochondrial Ca^2+^ Handling

Besides the ER, mitochondria also provide additional Ca^2+^-buffering capacity and intervene in multiple Ca^2+^-mediated signaling processes [25]. The Ca^2+^ uptake in the mitochondria is driven by the negative mitochondrial membrane potential (−180 mV) and is facilitated by the voltage-dependent anion channel (VDAC) on the outer mitochondrial membrane and the mitochondrial calcium uniporter (MCU) on the inner mitochondrial membrane. The Ca^2+^ is removed from the mitochondrial matrix via the Na^+^/Ca^2+^ exchanger (NCLX), located on the inner mitochondrial membrane. The NCLX transports 3Na^+^ per Ca^2+^, but also displays Li^+^-dependent Ca^2+^ transport [26]. The NCLX is regulated by kinase activity, mitochondrial membrane potential and proteases [26,27].

The ER is in close contact with the mitochondria, focalized at the mitochondria-associated ER membranes (MAMs) [28]. The MAMs are dynamic structures that allow ‘quasi-synaptic’ Ca^2+^ transfer from the ER to the mitochondria. The distance between the ER and the mitochondria at these MAMs varies from 10 nm to 30 nm [29,30].

In the MAMs, the IP_3_Rs associate in a macromolecular complex with VDAC1 and GRP75 [31]. As such, the local elevations of [Ca^2+^] in the microdomains, which can reach more than 10 μM Ca^2+^, can overcome the inherently low Ca^2+^ affinity of MCU, ensuring an adequate transfer of Ca^2+^ into the mitochondrial matrix, even under basal conditions [32]. Additionally, the translocase of the outer membrane 70 (TOM70) has been shown to bind to the IP_3_R and to facilitate the IP_3_R-mediated Ca^2+^ transfer from the ER to the mitochondria [33]. Besides TOM70, other accessory proteins aid in MAM formation and stabilization, including the FUN14 domain-containing protein 1 (FUNDC1) and inositol-requiring enzyme 1 α (Ire1α) [34,35]. Although all of the IP_3_R isoforms are found in the MAMs, IP_3_R type 2 appears to be the most efficient at transferring Ca^2+^ from the ER to the mitochondria [36]. Importantly, the intracellular [Ca^2+^] plays a dual role in regulating cell death [37]. Pro-survival Ca^2+^ oscillations promote the mitochondrial bio-energetics and the production of ATP by enhancing the activity of Ca^2+^-dependent enzymes (pyruvate dehydrogenase, isocitrate dehydrogenase and α-ketoglutarate dehydrogenase) [3]. Instead, sustained Ca^2+^ elevations trigger cell death [28,38] involving a Ca^2+^-dependent dissociation of cardiolipin from complex II, leading to excessive ROS production and the opening of the mitochondrial permeability transition pore (mPTP) [39,40].

The neurons have limited glycolytic capacity and therefore rely on proper mitochondrial ATP production for the maintenance of ionic gradients and for overall synaptic function [41]. The importance of MAM integrity in neurons is underpinned by the presence of the ER chaperone sigma-1 receptor (S1R) in the MAMs, where it stabilizes IP_3_R and enhances the ER–mitochondrial Ca^2+^ transfer [42]. The S1R is implicated in neuroprotection and neuroplasticity and is bound to the lumenal ER chaperone binding immunoglobulin protein (BiP). Under the conditions of [Ca^2+^] ER depletion, S1R dissociates from BiP and loses its specific MAM localization [42].

Another example of a MAM protein regulating the ER–mitochondrial Ca^2+^ transfer is ER oxidoreductase 1-α (Ero1α), for which tightly regulated expression levels are necessary to ensure proper mitochondrial Ca^2+^ signaling. On the one hand, the knockdown of Ero1α inhibits ER–mitochondrial Ca^2+^ transfer by impacting both the IP_3_R-mediated Ca^2+^ release and mitochondrial Ca^2+^ uptake. On the other hand, the overexpression of Ero1α leads to elevated Ca^2+^ basal efflux through the IP_3_R, lowering the steady state ER [Ca^2+^] and reducing agonist-induced ER–mitochondrial Ca^2+^ transfer [43].

Besides regulating Ca^2+^, the MAMs are also involved in redox regulation by functioning as hosts for the sources and targets of reactive oxygen species (ROS) [44]. Interestingly, the Hajnóczky lab developed a novel strategy for measuring ROS at the MAMs by using drug-inducible synthetic linkers to target fluorescent H_2_O_2_ sensors and ROS producers [44]. In doing so, they discovered dynamic H_2_O_2_ nanodomains present at the MAMs that are regulated by Ca^2+^ signaling. Moreover, it was shown that mitochondria generate oxidative bursts that are sensed by IP_3_R, which in turn respond by generating Ca^2+^ signals [45,46].

### 2.3. Lysosomal Ca^2+^ Handling

Besides the plasma membrane and mitochondria, the ER is also in close contact with other cellular organelles, such as lysosomes. In fact, the lysosomes specifically buffer the cytosolic [Ca^2+^] elevations established by IP_3_R-mediated Ca^2+^ release, but not the [Ca^2+^] elevations caused by other processes, such as SOCE [47]. Additionally, several Ca^2+^-release channels are embedded in the lysosomal membrane, i.e., transient receptor potential mucolipin (TRPML), two-pore channel 2 (TPC2) and P2X4 receptors [48]. At the ER–lysosomal contact sites, these lysosomal Ca^2+^ release channels may activate the ER Ca^2+^ release channels by CICR [49]. In the case of TPC2-mediated Ca^2+^ release, nicotinic acid adenine dinucleotide phosphate (NAADP) seems to be involved as a second messenger, by binding the Jupiter microtubule-associated homolog 2 (JPT2) and like-Sm protein 12 (LSM12), which were identified as interactors with TPC2 [50,51,52]. While TPC2 receptors function as Ca^2+^ permeable channels activated indirectly by NAADP, TPC2 receptors are directly activated by phosphatidylinositol 3,5 bisphosphate (PI(3,5)P2), activating TPC2 to selectively flux Na+ ions [53]. Additionally, the NAADP-producing enzymes, NADPH oxidase (NOX) 1 (NOX1)/NOX2 and dual NADPH oxidase (DUOX) 1 (DUOX1)/DUOX2, have been identified [54].

## 3. Alzheimer’s Disease

Over 55 million people worldwide live with dementia; a number that is rising on a daily basis due to aging of the global population [55]. Alzheimer’s disease (AD) accounts for around 60–70% of these cases and is therefore the most frequently occurring form of dementia. Patients with AD clinically present with behavioral changes, memory impairment, difficulties with communication and troubled reasoning and judgement [56]. The major neuropathological hallmarks of the disease include the irreversible loss of neurons, the formation of intracellular neurofibrillary tangles consisting of hyperphosphorylated tau protein and the accumulation of extracellular amyloid beta (Aβ) aggregates [57]. These Aβ peptides are formed by the subsequent proteolytic cleavage of the amyloid precursor protein (APP) by α-, β- and γ-secretases. The latter are mainly responsible for the generation of the Aβ peptides with a length of 42 amino acids (Aβ42), which are more prone to form aggregates and to accumulate in the brain than the more frequently observed Aβ40 [58,59]. An additional hallmark of AD is the chronic activation of microglial cells, resulting in increased neurotoxicity [60].

The more rare, familial form of AD (FAD) is characterized by an early-onset of the disease and is caused by autosomal dominant mutations in APP or in presenilins, the catalytic subunits of the γ-secretase complex. Though familial AD occurs less frequently than the sporadic, late-onset form of the disease, the pathological features of both forms are similar [61]. The major risk factors for the development of sporadic AD are age, gender—given that two-thirds of people with AD are female [55]—and carrying at least one apolipoprotein E (APOE) ε4 allele [62,63].

There is currently no disease-modifying therapy available for AD, as the only available treatment focusses on symptom relief. The current therapy consists of administration of three different acetylcholinesterase inhibitors (donepezil, rivastigmine and galantamine) and the NMDA receptor antagonist, memantine. Previous anti-amyloid strategies developed in the past few decades were limited in their success [64]. The more recently FDA-approved aducanumab bears more promise but awaits further confirmation of clinical benefit [65]. In addition, the neuroprotective properties of S1R have prompted several researchers to assess the potential of pharmacological S1R agonists (such as choline and PRE-084) in AD models. Choline ameliorated the amyloidogenic processing of APP and reduced the AD-related microglial activation in an AD mouse model [66]. PRE-084 reduced the Aβ levels in the brain and ameliorated the learning and memory deficits of mice injected with Aβ [67].

### 3.1. From Amyloid Hypothesis to Calcium Hypothesis

In 1992, Hardy and Higgins hypothesized that amyloid beta proteins were the causative agent in the pathology of AD [68]. To date, the Aβ hypothesis is the most widely accepted hypothesis and has served as the main guide for the development of therapeutic strategies. Although much evidence to support this hypothesis has been gathered over the past years [69], the hypothesis remains somewhat controversial [70]. One of the reasons to cast doubt upon the Aβ hypothesis is that the formation of Aβ plaques is also observed in cognitively well-functioning individuals and that the overall correlation between plaque deposition and cognitive function is rather weak [71,72,73,74]. Additionally, the failure of many anti-Aβ therapies suggest that amyloid plaque formation is not the only pathogenic factor involved in AD [64].

Only a few years later, Zaven S. Khachaturian proposed the Ca^2+^ hypothesis of AD [75]. This hypothesis is based on the similar pathological processes observed in AD and aging, such as increased oxidative and metabolic stress, decreased ATP production and overall Ca^2+^ dysregulation. In the following years, more and more evidence to support the Ca^2+^ hypothesis emerged and emphasized the important role of altered Ca^2+^ signaling in the pathophysiology of AD [76,77]. Moreover, targeting the dysregulation in Ca^2+^ homeostasis has been suggested as a potential approach for the treatment of AD [77,78]. The various downstream effects of dysregulated Ca^2+^ homeostasis include several molecular alterations underlying the AD pathogenesis, such as loss of synapses, mitochondrial dysfunction, oxidative stress and dysfunctional synaptic transmission and plasticity [79,80,81]. Of note, the alterations in Ca^2+^ handling are not exclusive for AD pathology, but have also been linked to other neurodegenerative diseases including Parkinson’s disease and Huntington’s disease [82].

Dreses-Werringloer et al. discovered a Ca^2+^-permeable channel named calcium homeostasis modulator 1 (CALHM1), directly linking Ca^2+^ homeostasis to AD and to the production of Aβ [83,84]. The CALHM1 is a transmembrane glycoprotein that generates a large Ca^2+^ conductance across the plasma membrane. The CALHM1 plays a role in neuronal excitability and shares similar properties with connexins, pannexins and innexins [85,86]. It was shown that a single nucleotide polymorphism in the CALHM1, resulting in proline substitution to leucin at codon 86, was associated with different AD cases. Moreover, this mutation was reported to cause an impairment of the plasma membrane Ca^2+^ permeability, a reduction in the cytosolic Ca^2+^ rise after removing extracellular Ca^2+^ and to affect APP processing [83].

Multiple studies indicate an increase in the resting cytosolic [Ca^2+^] in AD. Kuchibhotla et al. reported a Ca^2+^ overload in individual neurites and spines in mice, which was dependent on the proximity to Aβ deposits [87]. Moreover, Aβ plaque deposition was required to induce Ca^2+^ overload. The magnitude of the Ca^2+^ overload was similar to levels that are associated with long-term depression, an important neuronal process characterized by decreased synaptic strength that is also affected in AD. Similarly, resting cytosolic [Ca^2+^] was significantly increased in the cultured, cortical neurons from a triple transgenic AD mouse model (3xTg-AD). However, in the absence of extracellular Ca^2+^, the intracellular Ca^2+^ levels almost completely returned to normal, suggesting that an influx of extracellular Ca^2+^ plays a role in the observed intracellular Ca^2+^ overload. It was further demonstrated that increases in Ca^2+^ influx via L-type Ca^2+^ channels and increased IP_3_R-mediated Ca^2+^ release are responsible for the elevated cytosolic Ca^2+^ levels. The observed disturbances in the Ca^2+^ signaling occur prior to the onset of plaque formation and neurofibrillary tangles, indicating the crucial role of Ca^2+^ signaling in the early pathophysiology of AD [88].

### 3.2. Changes in ER Ca^2+^ Handling in AD

The Ca^2+^ hypothesis was further supported by the discovery of various FAD-linked mutations in PSEN causing alterations in intracellular Ca^2+^ release (Figure 2) [11,89,90]. Two possible mechanisms for this have been proposed. On the one hand, it is suggested that PSENs act as ER Ca^2+^ leak channels, independent of their function as a catalytic subunit of the γ-secretase complex, and that this leak function is abolished due to PSEN mutations [11]. Additionally, the ER Ca^2+^ overload observed in the cells expressing PSEN mutants, leads to increased Ca^2+^ release after IP_3_R activation. Interestingly, the PSENs undergo endoproteolytic cleaving in order to be stable and functional in the γ-secretase complex, but the authors suggest that the holoprotein of PSENs is responsible for the ER Ca^2+^ leak function. Indeed, the PSEN1 ΔE9 mutant, a mutant that is unable to undergo endoproteolytic cleavage, showed higher ER Ca^2+^-leak channel activity than WT PSEN1 [11]. Additionally, the phosphorylation of PSEN1 appeared critical for its ER Ca^2+^-leak function [10,91]. On the other hand, single channel analyses showed that different PSEN1 mutations directly sensitize IP_3_R activity, thereby resulting in hyperactive Ca^2+^ signaling in four different cell systems [90]. This gain-of-function effect of FAD-linked PSEN mutations was also independent of the γ-secretase activity of PSEN. More specifically, the PSEN mutations drive the IP_3_R into the H mode, the gating mode with high open probability and bursting channel behavior [92], causing a sufficiently large Ca^2+^ flux to enable signal amplification by CICR [90]. In agreement, Stutzmann et al. showed increased IP_3_-evoked Ca^2+^ responses in neurons expressing PSEN1 mutants by using whole-cell patch-clamp recording, flash photolysis of the caged IP_3_ and two-photon imaging in brain slices [93]. In addition, administration of oligomeric Aβ42 was observed to raise cytosolic [Ca^2+^] through an ER Ca^2+^ release, in part via an IP_3_R-mediated Ca^2+^ release, but also through an IP_3_R-independent Ca^2+^ leak from the ER [94].

Besides the IP_3_R, changes in the RyR activity have also been linked to AD (Figure 2). Chan et al. demonstrated an increased expression of the RyR3 in cultured neurons harboring PSEN1 mutations. Consequently, Ca^2+^ release following pharmacological stimulation of RyR was enhanced in comparison to WT PSEN1 [95]. Similar results were obtained in the 3xTg mouse model of AD [96]. From co-immunoprecipitation studies, a direct interaction between PSEN1 and RyR was suggested. However, it has not yet been determined how the PSEN mutants would affect interaction with RyR [95]. With the use of radiolabeled ryanodine, an increase in the RyR expression in the subiculum and in the CA1 and CA2 regions of the hippocampus was demonstrated in the very early stages of pathology. In contrast, a reduced binding of radiolabeled ryanodine, and thus RyR loss, was observed in the later stages of the disease in all of the hippocampal regions that are affected by neurofibrillary pathology in AD [97]. Additionally, it was demonstrated that the IP_3_-evoked Ca^2+^ release occurs in the greater majority of the AD mice mediated through the RyR, which is not the case in the WT mice. This is probably due to CICR initiated by large Ca^2+^ releases through upregulated RyRs [98]. Importantly, the ER Ca^2+^ signaling, and, more specifically, the RyR-mediated Ca^2+^ release, plays a role in synaptic transmission. The enhanced RyR-mediated Ca^2+^ signals that are observed in the 3xTg-AD mice can therefore have a modulatory effect on the synaptic activity that are not present in the control mice. Moreover, the RyR-mediated Ca^2+^ release impacts both presynaptic and postsynaptic events in the 3xTg-AD mice, but plays a much smaller role in the control mice [99].

The neurons derived from transgenic mice expressing mutant PSEN1 also displayed a decrease in SOCE, which is independent of the expression of APP and independent of the γ-secretase activity of PSEN1 [100]. Additionally, Yoo et al. indicated that treating cells with a SOCE inhibitor selectively increased the generation of Aβ42 peptides, suggesting that impairments in SOCE may be an early event in AD pathology [101]. On the other hand, an enhancement in the ER [Ca^2+^] was observed which did require APP expression [100]. Similar elevated ER Ca^2+^ levels were found in the fibroblasts carrying a FAD-linked PSEN1 mutation. It is suggested that this elevation in the ER Ca^2+^ levels is responsible for the attenuation of SOCE, since reaching low Ca^2+^ levels below the threshold to activate SOCE is prevented [102]. The exact mechanism by which the PSEN1 mutations cause this increase in intracellular Ca^2+^-store filling is not yet elucidated. However, previous research indicated an increased expression of acylphosphatase, an enzyme enhancing the activity of SERCA and thus supporting ER-store filling, in the fibroblasts of patients with AD [102,103].

Green et al. further demonstrated that the PSENs play a role in ER store filling via SERCA, as it was reported that the PSENs are necessary for properly maintaining SERCA activity [104]. Moreover, the PSENs were shown to colocalize and physically interact with SERCA. The knockdown of SERCA gave rise to a phenotype resembling presenilin-null cells, characterized by increased basal cytosolic Ca^2+^ levels, which could be rescued by overexpressing PSEN. Additionally, SERCA was demonstrated to impact on the Aβ production with the inhibition of SERCA activity rapidly reducing the production of Aβ40 and Aβ42 proteins [104].

To conclude, an important role of Ca^2+^ signaling in early AD is emerging. A vicious feed forward cycle appears to exist between Ca^2+^-signaling dysregulation and APP processing [105] (Figure 3). On the one hand, aberrant Ca^2+^ signals have an effect on APP processing [106] although studies researching this topic are rather sparse. On the other hand, administration of oligomeric Aβ was shown to increase cytosolic [Ca^2+^] [94], thereby further driving the formation of Aβ.

### 3.3. Mitochondrial Ca^2+^ Dysregulation in AD

It has become more and more clear that mitochondria play a fundamental role in the pathophysiology of AD (Figure 4) [107]. More specifically, mitochondrial dysfunction has been reported as an early event in AD, as it precedes the onset of plaque formation in the 3xTg-AD mouse model [108]. Yao et al. demonstrated decreased mitochondrial bioenergetics, increased oxidative stress and increased mitochondrial amyloid load in the 3xTg-AD mouse model [108]. Other studies confirm the impairment in mitochondrial function in the brains of AD mouse models or AD patients [109,110,111].

The mitochondrial dysfunction observed in AD is clearly linked to the occurrence of Aβ deposits as Aβ has been detected in the mitochondria of AD mouse models and postmortem brains of AD patients [112]. It is suggested to be imported in the mitochondrial matrix via the translocase of the outer mitochondrial membrane (TOM) machinery [113], and the accumulated Aβ directly impairs the mitochondrial Ca^2+^ homeostasis. Mitochondrial Ca^2+^ overload was detected in an AD mouse model, but only after the appearance of Aβ in the brains of the mice, suggesting the importance of Aβ deposits as the initial trigger for this mitochondrial Ca^2+^ overload. Moreover, Calvo-Rodriguez et al. demonstrated that the application of soluble Aβ oligomers to the brain of naïve mice caused an increased mitochondrial Ca^2+^ uptake and elicited subsequent mitochondrial Ca^2+^ overload [114]. Mitochondrial Ca^2+^ overload consequently results in severe oxidative damage, loss of mitochondrial membrane integrity, deprivation of ATP production and eventually cell death [115].

The exact mechanism by which Aβ deposits provoke mitochondrial Ca^2+^ uptake has not been established. It has been suggested that Aβ forms Ca^2+^-permeable channels in bilayer membranes [116], or that it causes the formation of non-specific ion channels in the plasma membrane and the membranes of intracellular organelles [117]. Alternatively, increased Ca^2+^ uptake via the MCU complex is thought to play a role [118]. The selective blocking of the MCU complex abolished the mitochondrial Ca^2+^ overload caused by soluble Aβ application, indicating the potential of the MCU complex as a therapeutic target [114,119]. Another proposed mechanism for the mitochondrial Ca^2+^ overload is via the impaired Ca^2+^ efflux through the Na^+^/Ca^2+^ exchanger (NCLX), the main mitochondrial Ca^2+^ efflux transporter. The brain samples from AD patients showed a reduced expression and functionality of the NCLX, causing an increase in the mitochondrial Ca^2+^ concentration [120,121]. The genetic rescue of NCLX expression in 3xTg-AD mice abolished the cognitive decline and reduced neuronal pathology. Moreover, the mitochondrial Ca^2+^ efflux via the NCLX is also inhibited by tau, thereby linking the other major pathophysiological hallmark of AD to impaired mitochondrial Ca^2+^ handling [122].

A direct molecular link between Aβ and mitochondrial dysfunction is established by Aβ-binding alcohol dehydrogenase (ABAD), a member of the short-chain dehydrogenase-reductase family that is enriched in the mitochondria of neurons. The ABAD-protein levels were reported to be elevated in the affected regions of the AD brain [123]. Moreover, ABAD and Aβ form a highly specific complex that is detected in the brains of AD patients and in the mitochondria of a transgenic AD mouse model [124]. Similarly, Yao et al. described a correlation between the increase in mitochondrial Aβ and the rise in ABAD levels in the brains of 3xTg AD mice [108]. The ABAD-Aβ complexes elicit cellular dysfunction by causing ROS leakage, mitochondrial dysfunction, increased opening of the mitochondrial membrane permeability transition pore (MPT) and eventually cell death [123].

### 3.4. MAMs and AD

MAM homeostasis is key for cellular health [28] and is well-established to be affected in neurodegenerative diseases, including AD. Moreover, altered MAM function is even suggested as a main pathogenic cause in AD [125]. The evidence to support the role of MAMs in AD pathophysiology comes from the observation that APP, PSENS and the complete γ-secretase complex predominantly reside in the MAMs [126]. Additionally, the ER–mitochondrial communication and MAM function is increased in AD patient fibroblasts and in AD animal models [127]. Fernandes et al. reported that cells overexpressing the Swedish mutation in APP cause an accumulation of APP in MAMs and mitochondria and affect the ER-mitochondria contacts, ER-mitochondria Ca^2+^ transfer and mitochondrial function and dynamics [125]. Moreover, the exposure of hippocampal neurons to Aβ deposits triggered an increase in the number of the ER–mitochondrial contact sites and enhanced the expression of different MAM-associated proteins, including IP_3_R3 and VDAC1. Similarly, an upregulation of MAM-associated proteins was also detected in the APPSwe/Lon mouse model, which overexpresses the Swedish (K670N/M671L) and London (V717I) mutations in APP [128].

Alternatively, C99, a product derived from APP cleavage by β-secretase, accumulates at the MAMs and causes disturbances in the sphingolipid homeostasis and overall alterations in the lipid composition of the MAMs. Therefore, this MAM-localized C99 accumulation has been reported as a driver of the mitochondrial dysfunction in AD [129]. Moreover, C99 accumulation triggers mitochondrial dysfunction independent of Aβ in both in vitro and in vivo models of AD [130]. In agreement, Lauritzen et al. found that C99 accumulated at early stages in the 3xTg AD mice, months before the Aβ deposits are detected. This accumulation is particularly present in AD-sensitive brain regions, such as the hippocampus [131].

## 4. Wolfram Syndrome

### 4.1. Introducing Wolfram Syndrome

Wolfram syndrome (WS) is a rare, genetic neurodegenerative disease first described by Wolfram and Wagner in 1938 [132]. The main pathological hallmarks of WS include optic nerve atrophy, diabetes insipidus, diabetes mellitus and hearing loss [133]. Additionally, WS patients often present with different neurological conditions, such as ataxia, brain stem atrophy, autonomic and peripheral neuropathy, headaches and seizures [134]. The treatment is mainly limited to ameliorating the emerging symptoms and patients usually die around the age of 30, often due to respiratory failure caused by brain stem atrophy [135].

Two different types of WS exist, namely WS type 1 (WS1), accounting for over 90% of the cases, and WS type 2 (WS2). The pathophysiology of WS1 and WS2 slightly differs, as diabetes insipidus is solely observed in the WS1 patients, while upper intestinal ulcers and defective platelet aggregation are mainly present in patients with WS2 [136,137]. Patients with WS1 carry a loss-of-function mutation in the WFS1 gene encoding for wolframin (Wfs1), a transmembrane protein located at the ER. The Wfs1 is highly expressed in brain tissue, pancreatic β-cells and in the heart [138,139,140]. Mutations in WFS1 cause elevations in ER stress, alterations in pancreatic β-cells and cause stress-induced apoptosis [141]. Moreover, wolframin has been shown to play a role in maintaining ER homeostasis and in the unfolded protein response (UPR) [142,143,144].

CISD2 is the second causative gene for WS, encoding a 15 kDa redox active protein named Cisd2, though known under various aliases, such as Miner1, Naf-1, ERIS, WS2 and CDGSH2 [145]. Cisd2 is an ER-localized protein, but is enriched in the MAMs [146]. The physiological roles of Cisd2 are not completely elucidated, but Cisd2 is thought to be involved in mitochondrial iron transport [147]. Through the CDGSH domain, Cisd2 can bind to iron–sulfur clusters, which can be donated by Cisd1, a close family member of Cisd2 [148]. In this way, Cisd1-Cisd2 might partake in a relay system responsible for shuttling iron outside of the mitochondria [147]. Moreover, Cisd2 is generally viewed as a prolongevity factor and therefore plays a role in aging and various cancers. Most cell types ubiquitously express Cisd2, underscoring its vital role in maintaining cellular homeostasis [149]. For instance, the overexpression of Cisd2 in an APP/PS1 transgenic mouse model for AD resulted in a neuroprotective effect, probably by preventing mitochondrial damage [150]. Conversely, a lack of Cisd2 can lead to chronic injury, such as that seen in the cornea of patients with epithelial disease [151]. A vicious cycle of aberrant wound healing, hindered by Ca^2+^ dyshomeostasis and mitochondrial dysfunction, leads to the hyperproliferation and exhaustion of the corneal epithelial stem cells [151,152].

### 4.2. The Role of Ca^2+^ in WS

A unifying principle for the two distinct WS-associated proteins, Wfs1 and Cisd2, is their involvement in Ca^2+^ homeostasis (Figure 5) [145]. An early electrophysiological study of Xenopus oocytes already pointed towards an interplay between the Wfs1 and IP_3_Rs. The channel activity recordings in the lipid bilayers uncovered an elevated IP_3_-induced current when Wfs1 is reconstituted into the bilayers [153]. Of note, the authors hypothesized that Wfs1 can function as a Ca^2+^-permeable channel [153], though no further studies validated this. Conversely, the Wfs1-deficient neurons displayed decreased IP_3_R-mediated Ca^2+^ release and perturbed mitochondrial dynamics, leading to delayed development [154]. Further evidence for a stimulatory role for Wfs1 on IP_3_R-mediated Ca^2+^ release stems from a study of WS1 patient fibroblasts, which showed that Wfs1 is important for the Neuronal Ca^2+^ Sensor 1 (NCS1) stability and binding to IP_3_R in the MAMs. Since NCS1 can sensitize IP_3_R to IP_3_, the Wfs1 stimulates the ER–mitochondrial Ca^2+^ transfer [155]. A second member of the Ca^2+^ toolkit interacting with Wfs1 is the SERCA pump. The Wfs1 was reported to bind and regulate SERCA abundance by mediating its proteasomal degradation [156].

The exact impact of Wfs1 on cellular Ca^2+^ homeostasis is still not entirely clear, but most studies seem to point towards a higher resting cytosolic [Ca^2+^] upon Wfs1 deficiency, possibly through an increased Ca^2+^ leak from the ER through the Ca^2+^-release channels [154,157,158]. Additionally, the reduced ER Ca^2+^ uptake could explain the elevated cytosolic Ca^2+^ levels, since in the Wfs1-deficient cells, the ER Ca^2+^ store content and store refilling is decreased [159]. These elevated cytosolic Ca^2+^ levels are thought to hyperactivate calpain 2, a Ca^2+^-dependent pro-apoptotic protease. In vitro therapeutic interventions with dantrolene, an RyR inhibitor, normalized the resting cytosolic [Ca^2+^] and prevented excessive calpain 2-mediated cell death in the Wfs1-deficient cells [157]. Similarly, the calpain inhibitor XI and ibudilast, a known interactor with NCS1, ameliorated the resting cytosolic [Ca^2+^] and excessive cell death in the Wfs1-deficient cells [158]. Currently, the administration of dantrolene to WS patients is being investigated in a clinical trial. Generally, dantrolene seems to be well-tolerated by WS patients; however, the efficacy of dantrolene as a disease-delaying or -reverting compound is still unknown and awaits confirmation in a trial with a larger sample size. In pediatric patients, a modest increase in β cell function could be observed, which might indicate a certain level of remaining β cells is necessary for dantrolene efficacy [160].

Similar to Wfs1, a deficiency in Cisd2 leads to elevated cytosolic Ca^2+^ levels [146,161,162,163,164]. Cisd2 also interacts with IP_3_R as well as B-cell lymphoma 2 (Bcl-2), a known modulator of IP_3_R [165,166]. This raises the possibility that Cisd2 can indirectly regulate IP_3_R activity through Bcl-2 [145]. In fact, biophysical studies have established that the Cisd2-Bcl-2 complex relies on the interaction between the catalytic domain of Cisd2 with the BH4 domain of Bcl-2 [167], the very same domain that is responsible for IP_3_R inhibition [168,169]. Alternatively, CISD2 might directly modulate IP_3_R activity, irrespective of Bcl-2 [145]. The IP_3_R activity is strongly impacted by oxidation [46,170] including at the ER–mitochondrial contact sites, where the ROS boost the IP_3_R-mediated Ca^2+^ fluxes [45]. As Cisd2 seems to function as a cellular antioxidant, cells lacking Cisd2 might be prone to increased IP_3_R oxidation [149,171].

An additional mode of action of Cisd2 in the regulation of cytosolic [Ca^2+^] could occur via modulating SERCA activity. Indeed, findings from murine Cisd2 KO fibroblasts [172], hepatocytes from heterozygous Cisd2 KO mice [173] and cardiomyocytes from Cisd2 KO mice [162] suggest that ER [Ca^2+^] is decreased when lacking Cisd2. Additionally, Cisd2 interacted with SERCA2b [173] and SERCA2a [162]. On the contrary, in WS2 patients’ lymphoblastoids [174] and Cisd2 KO murine myoblasts [164], the ER [Ca^2+^] was increased, while the findings in other cell types reported unchanged ER [Ca^2+^] upon loss of Cisd2 [164,165]. Although it is clear that Cisd2 impacts on the Ca^2+^ homeostasis, much remains to be discovered on how Cisd2 interacts with the different actors in the Ca^2+^ toolkit.

### 4.3. Mitochondrial Dysfunction in WS

Although WS was initially considered to be a mitochondriopathy, the research focus has shifted towards ER dysfunction and ER–mitochondrial crosstalk, due to the presence of both the Wfs1 and Cisd2 in the MAMs [133]. For instance, the Wfs1 was shown to partake in a macromolecular complex in the MAMs involving IP_3_R and NCS1. The Wfs1-deficient cells exhibited reduced ER–mitochondrial contact and ER–mitochondrial Ca^2+^ transfer, ultimately resulting in impaired mitochondrial function [155,158]. At the MAMs, Wfs1 interacts with S1R, which similarly to NCS1 can also positively modulate ER–mitochondrial Ca^2+^ transfer. Stimulation of S1R with PRE-084, a S1R agonist, could overcome the Wfs1-deficiency-mediated decrease in IP_3_R-mediated Ca^2+^ release and ER–mitochondrial Ca^2+^ transfer [175]. The PRE-084, could also alleviate mitochondrial abnormalities associated with reduced MAM functionality, such as increased mitophagy [175]. Additionally, WS1 is linked with impaired mitochondrial dynamics, as knocking down the Wfs1 in neurons decreased the mitochondrial mass, mobility and fusion, while increasing the rate of mitophagy [154].

Similarly, Cisd2 was established to be a MAM-resident protein. In the MAMs, Cisd2 can interact with the protein GTPase IMAP family member 5, which is vital for the differentiation process of adipocytes [146]. Moreover, a study of the WS2 patients’ fibroblasts found an increase in the ER–mitochondrial Ca^2+^ transfer and closer ER–mitochondrial contact, leading to mitochondrial hyperfusion [146]. Furthermore, the absence of Cisd2 leads to impaired biogenesis [146], aberrant mitochondrial metabolism [176], increases in mitochondrial ROS [177] and iron levels [178,179] and mitochondrial swelling [164].

Of note, the deficiencies in both Wfs1 or Cisd2 are associated with ER stress [139,142,172,180,181], a process which by itself is known to lead to mitochondrial dysfunction [182].

## 5. Similarities between AD and WS

AD and WS are both neurodegenerative disorders that display very different clinical manifestations. However, some similarities can be found between these two distinct diseases. A first resemblance between AD and WS is that no disease-modifying treatment is available, and the only available therapy is focused on ameliorating the symptoms. This points out the difficulties of generating therapeutic strategies in neurodegeneration and already shows that much more research is needed in the field.

Mechanistically, some other similarities can be observed, mainly in the early pathology of the diseases. A first link between the two diseases involves tau. The hyperphosphorylated tau forming neurofibrillary tangles is a major hallmark of AD and is serving as a potential therapeutic target [183]. Recently, Chen et al. described a novel function for Wfs1 in the development and progression of tau pathology [184]. The Wfs1 can interact with tau protein and affect its aggregation and propagation. Moreover, increased Wfs1 reduced tau pathology and neurodegeneration in PS19 mice, a widely used tauopathy model which resembles AD-like pathology. On the other hand, Wfs1 deficiency increased the pathological tau and apoptosis, and impaired the spatial learning and memory in PS19 mice [184].

Most of the similarities between AD and WS are related to Ca^2+^ homeostasis and mitochondrial function, indicating the pivotal role of (mitochondrial) Ca^2+^ signaling in the early stages of neurodegenerative disorders. The Ca^2+^ homeostasis plays a key role in AD, as indicated by the Ca^2+^ hypothesis of AD pathology [75,79], as well as in WS, where it is the unifying factor between Wfs1 and Cisd2 [145]. A common observation in both of the diseases is an increase in cytosolic [Ca^2+^] [87,90,93,154,157,158]. Different mechanisms for this have been proposed, but in both AD and WS, there is a link with the ER store filling via SERCA. The PSENs in AD and Wfs1 and Cisd2 in WS have been shown to interact with SERCA [102,103,104,156,162]. The Ca^2+^ overfilling of the ER can lead to exaggerated IP_3_R-mediated Ca^2+^ releases and thus contributes to elevated cytosolic [Ca^2+^]. Moreover, a direct link with the IP_3_R is established in both of the disorders. The mutations in PSEN have been shown to augment the activity of the IP_3_R [90], and both Wfs1 and Cisd2 display a direct and indirect interaction with the IP_3_R [145,149,154,155,171]. Additionally, hyperactive RyR-mediated Ca^2+^ release is linked with both WS and AD [157,185,186]. In the case of WS, the administration of dantrolene, an RyR inhibitor, can rescue this defect in vitro and its efficacy in vivo is currently being investigated in clinical trials [157,160]. Therefore, the use of dantrolene, either alone or in combination with immunotherapy, might also be considered for the treatment of AD. Additionally, post-translation modification of the RyR2 or RyR2 mutations have been linked to AD [186,187]. More specifically, Yao et al. reported that mice carrying the RyR2-R4496C mutation, which increases the open probability of the channel, displayed neuronal hyperactivity and impaired learning and memory formation. Similarly, increasing the open probability of RyR2 in 5xFAD mice exacerbated the onset of these symptoms [187]. Hence, it would be interesting to assess whether patients with RyR2 mutations or other channelopathies are at risk of not only developing AD but also exhibiting features that resemble WS. However, such RyR2 mutations have not yet been reported in WS patients.

The resemblances between AD and WS are not limited to the ER, but are also observed at mitochondrial level. The mitochondrial Ca^2+^ overload has been reported, as well as an increased ER–mitochondrial Ca^2+^ transfer in WS2 patients’ fibroblasts [146]. Similarly, the ER–mitochondrial Ca^2+^ transfer is affected in cells carrying mutations in APP [125]. In both AD and WS, this leads to mitochondrial dysfunction and increases in ROS production [108,109,110,111,155,156,157,158,177].

Another very interesting link between AD and WS is through S1R. S1R resides at MAMs where it acts on IP_3_Rs and regulates the Ca^2+^ signaling and survival [42]. The WFS1 was reported to interact with the S1R and therefore to positively modulate the ER–mitochondrial Ca^2+^ transfer [175]. Stimulation with a S1R agonist could counteract the impaired Ca^2+^ homeostasis and mitochondrial deficits observed in WS [175]. However, also in AD, S1R is emerging as a novel target. The S1R is downregulated in AD and other neurodegenerative diseases, although the exact mechanism for this downregulation is not understood [188,189,190]. The treatment of primary hippocampal cultures obtained from WT mice with an S1R agonist protected against the Aβ_42_ oligomer toxicity in mushroom spines [191]. Moreover, the S1R agonist also prevented mushroom spine loss in AD-causing mutations in hippocampal neuronal cultures [191]. Therefore, S1R stimulation might be an exciting strategy in both AD and WS, and in neurodegeneration in general, to target early defects found in the disease pathology.

Alternatively, the antiapoptotic protein Bcl-2 can also be linked to both diseases. Multiple members of the Bcl-2 family have been associated with AD, but mainly a downregulation of Bcl-2 itself has been linked to AD [192]. Moreover, Bcl-2 is suggested as a potential therapeutic target for AD since the overexpression of Bcl-2 in a transgenic AD mouse model suppressed the formation of plaques and neurofibrillary tangles and improved memory retention [193]. On the other hand, Cisd2 has been shown to directly interact with Bcl-2, and is important for Bcl-2′s function in inhibiting Beclin-1-mediated autophagy [165,166].

Finally, Liangping et al. reviewed the role of Wfs1 and Cisd2 in AD, providing a direct link between WS and AD [194]. For instance, Cisd2 overexpression was shown to promote survival and protect against neuronal loss in an AD mouse model, while Cisd2 deficiency accelerates AD pathology [150]. Therefore, Cisd2-based therapies could be exploited as potential therapeutic strategy in AD. Remarkably, Cisd2 also protects against mitochondrial damage and thus the protective effect to neuronal loss could also be mediated though the regulation of mitochondrial function [150].

In summary, WS and AD entail common hallmarks in early disease pathology that can be exploited as potential treatment strategies to target both diseases. The possibilities for such therapeutic targets include S1R, SERCA, RyR and the IP_3_R to normalize increased cytosolic [Ca^2+^], or Wfs1- and Cisd2-based therapies (Figure 6). It would be very beneficial to further investigate these similar targets so that potential drugs for AD can also be effective for WS, and vice versa.

## 6. Conclusions

AD and WS are both neurodegenerative diseases that display distinct hallmarks and a wide array of symptoms. However, both of the diseases in the early stages share some common hallmarks, including alterations in the ER and mitochondrial Ca^2+^ handling, mitochondrial dysfunction and increased ROS production. Disturbances in Ca^2+^ signaling are central in the early pathophysiology of AD, as well as in WS. This indicates that similar mechanisms may underly these two diseases’ pathologies and suggests that both research topics might benefit from complementary research.

## Figures and Tables

**Figure 1 cells-11-01963-f001:**
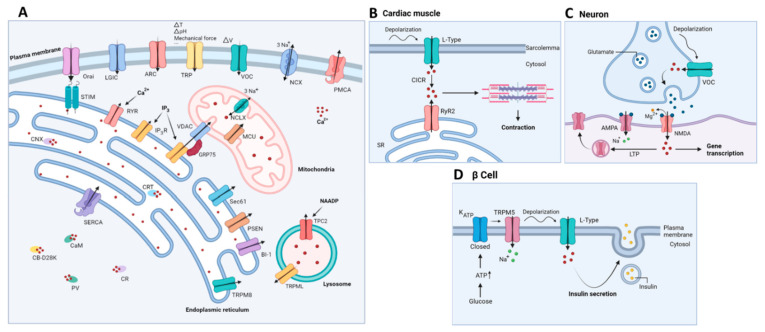
(**A**) Schematic representation of the Ca^2+^-signaling toolkit. Ca^2+^ can enter the cell via store-operated, STIM-gated Orai channels, but also through voltage-gated Ca^2+^ channels (VOC), arachidonic acid-regulated Ca^2+^ channels (ARC), transient receptor potential channels (TRP) and ligand-gated ion channels (LGIC). The plasma membrane Ca^2+^ ATPase (PMCA) and the plasmalemmal Na^+^/Ca^2+^ exchanger (NCX) are responsible for Ca^2+^ extrusion into the extracellular space. Ca^2+^ is mainly stored in intracellular Ca^2+^ stores, such as the endoplasmic reticulum, where it is buffered by calreticulin (CRT) and calnexin (CNX). Mobilization of Ca^2+^ from the ER into the cytosol occurs via two main channels, namely the inositol 1,4,5-trisphosphate receptor (IP_3_R) and the ryanodine receptor (RyR). Multiple ER-Ca^2+^ leak channels exist including presenilin (PSEN), Bak inhibitor-1 (BI-1), TRP melastin 8 (TRPM8) and Sec61 translocon. ER Ca^2+^ depletion is detected by luminal ER Ca^2+^ sensors STIM1/STIM2, which subsequently activate plasmalemmal Orai to cause a Ca^2+^ influx across the plasma membrane. (Re)filling of the ER is mediated by the sarco-endoplasmic reticulum ATPase (SERCA). In the mitochondria, Ca^2+^ uptake is mediated by the voltage-gated anion channel (VDAC) on the outer mitochondrial membrane and the mitochondrial calcium uniporter (MCU) complex on the inner mitochondrial membrane. The main mitochondrial Ca^2+^ efflux transporter is the Na^+^/Ca^2+^ exchanger (NCLX). Lysosomal Ca^2+^ release is mediated by transient receptor potential mucolipin (TRPML) and two-pore channel 2 (TPC2). This latter is regulated by nicotinic acid adenine dinucleotide phosphate (NAADP). Cytosolic Ca^2+^ is buffered by several Ca^2+^-binding proteins including calmodulin (CaM), calbindin D-28 (CB-D28K), calretinin (CR) and parvalbumin (PV); (**B**–**D**): Schematic overview of relevant examples of functional correlates of Ca^2+^ signaling in specific tissue/cell types. CICR, Ca^2+^-induced Ca^2+^ release; SR, sarcoplasmic reticulum; ER, endoplasmic reticulum; AMPA, α-amino-3-hydroxy-5-methyl-4-isoxazolepropionic acid; NMDA, N-methyl-D-aspartate; LTP, long term potentiation; TRPM5, transient receptor potential cation channel subfamily M member 5.

**Figure 2 cells-11-01963-f002:**
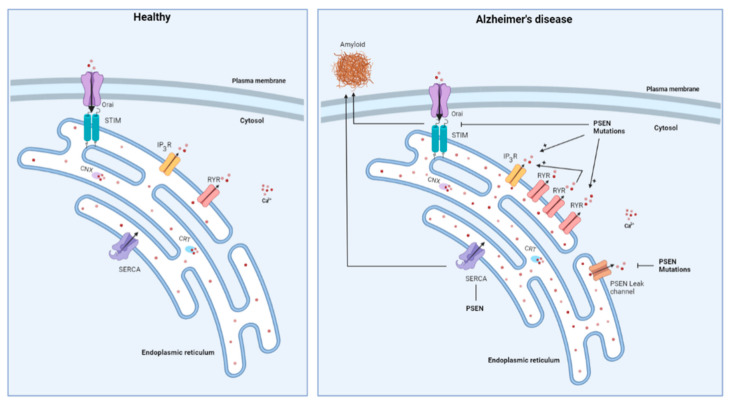
Schematic overview of changes in ER Ca^2+^ handling occurring in Alzheimer’s disease. Mutations in presenilin (PSEN) were shown to increase the expression of the RyR and to cause an enhancement of IP_3_R-mediated Ca^2+^ release. Additionally, the RyR is responsible for some of the IP_3_R-mediated Ca^2+^ release through CICR. PSENs were also reported to form ER Ca^2+^ leak channels, a function that is abolished by PSEN mutations and which causes ER Ca^2+^ overload. PSEN mutations also display an inhibitory effect on store-operated Ca^2+^ entry, which may be involved in the generation of toxic Aβ deposits. Finally, PSENs interact with SERCA which also impacts the Aβ formation.

**Figure 3 cells-11-01963-f003:**
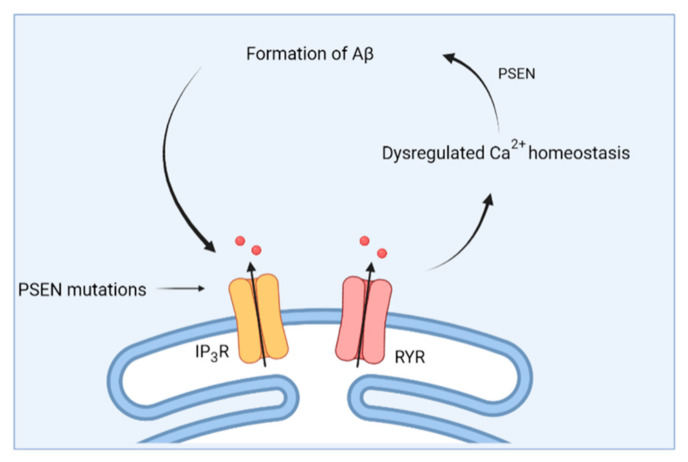
Schematic overview of the feed forward interplay between Ca^2+^ signaling and Aβ. Mutations in presenilin (PSEN) cause dysregulations in Ca^2+^ homeostasis, which then causes altered processing of APP and the formation of Aβ. In turn, this further impacts Ca^2+^ homeostasis.

**Figure 4 cells-11-01963-f004:**
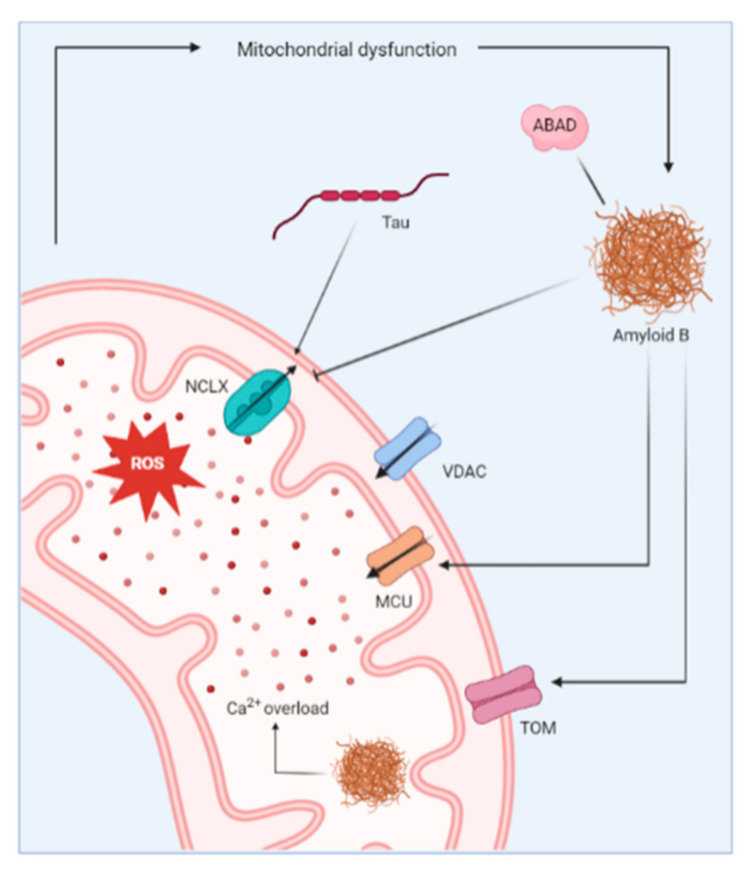
Representation of the mitochondrial changes reported in Alzheimer’s disease. Aβ has been shown to induce mitochondrial Ca^2+^ overload via different mechanisms. For instance, Aβ inhibits Ca^2+^ extrusion by blocking the Na^+^/Ca^2+^ exchanger (NCLX). On the other hand, Aβ promotes Ca^2+^ uptake via interaction with mitochondrial Ca^2+^ uniporter (MCU). Aβ deposits are also detected inside the mitochondria where they further contribute to dysregulated Ca^2+^ homeostasis. The uptake of Aβ in the mitochondria is reportedly regulated by translocase of the outer membrane (TOM) machinery. The mitochondrial Ca^2+^ overload causes increased production of ROS and further leads to mitochondrial dysfunction.

**Figure 5 cells-11-01963-f005:**
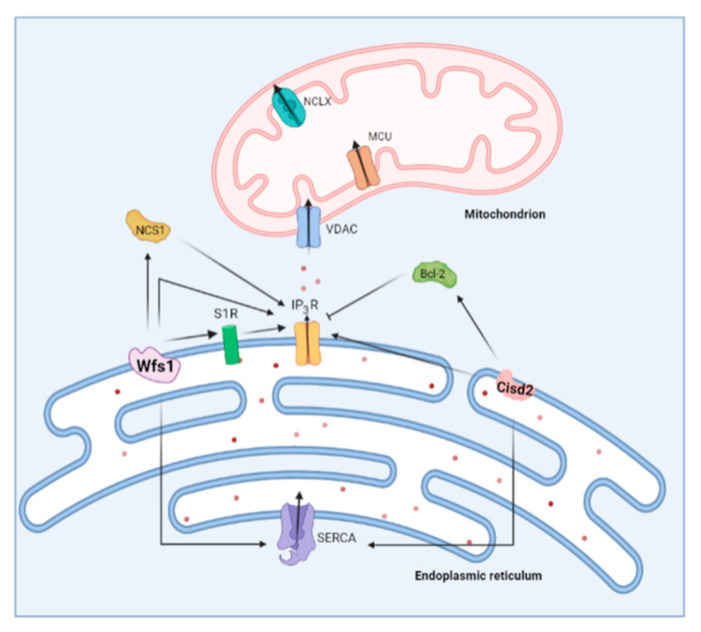
Overview of the role of two Wolfram syndrome proteins on ER Ca^2+^ handling. Wolframin (Wfs1) stimulates IP_3_R-mediated Ca^2+^ release both directly and via Neuronal Ca^2+^ Sensor 1 (NCS1). On the other hand, Cisd2 also interacts with the IP_3_R in a direct and indirect manner. Cisd2 modulates B-cell lymphoma 2 (Bcl-2), which is a known inhibitor of the IP_3_R. Both Wfs1 and Cisd2 were shown to interact with sarco-endoplasmic reticulum ATPase (SERCA).

**Figure 6 cells-11-01963-f006:**
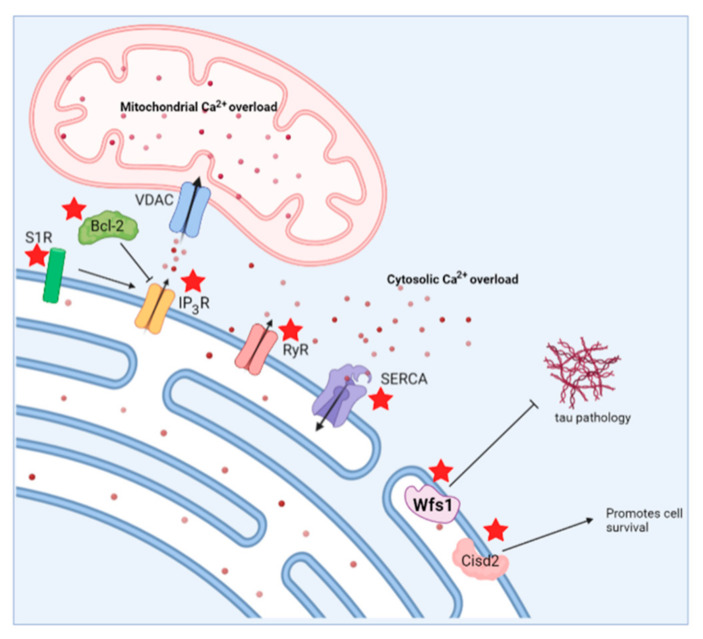
Schematic overview indicating potential therapeutic strategies to target both Alzheimer’s disease and Wolfram syndrome. Possible therapeutic targets are indicated by a red asterisk. S1R, sigma 1 receptor; Bcl-2, B-cell lymphoma 2; IP_3_R, inositol trisphosphate receptor; RyR, ryanodine receptor; SERCA, sarco/endoplasmic reticulum ATPase; VDAC, voltage-dependent anion channel; Wfs1, wolframin.

## Data Availability

Not applicable.

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
