# Peer review of "Dysregulated Ca2+ Homeostasis as a Central Theme in Neurodegeneration: Lessons from Alzheimer’s Disease and Wolfram Syndrome"

_cells, 2022, doi:10.3390/cells11121963_

Round 1

Reviewer 1 Report

Manon Callens et al wrote an interesting review on the common aspect of neurodegeneration in AD and WS. This hot topic is very important and suggests that the two diseases share common signaling pathway that are more similar than expected. The review is very well written and the figures are clear and in accordance with the text.

I have some suggestions to propose to the authors:

  • A specific paragraph may be added on Sigma-1 receptor and AD. This will help the reader when the Sigma-1 receptor and WS is described.
  • Another paragraph on AD and Tau may be added since there is a recent published paper demonstrating the link between Tau and WFS1 (Chen S et al. Acta Neuropathol. 2022 May;143(5):547-569). This link should be discussed before the conclusion.

Minor

Line 435, two references should be added on WS and ER stress:

 Sakakibara Y et al., Knockdown of wfs1, a fly homolog of Wolfram syndrome 1, in the nervous system increases susceptibility to age- and stress-induced neuronal dysfunction and degeneration in Drosophila. PLoS Genet. 2018 Jan 22;14(1):e1007196.

Crouzier et al., Morphological, behavioral and cellular analyses revealed different phenotypes in Wolfram syndrome wfs1a and wfs1b zebrafish mutant lines. Hum Mol Genet. 2022 Mar 21:ddac065. doi: 10.1093/hmg/ddac065.

Line 551, the reference 40 should be changed to 161.

Reviewer 2 Report

The focus of this review is to demonstrate calcium signaling as a unifying theme in two neurodegenerative disorders- Alzheimer's disease (AD) and Wolfram syndrome (WS).  The authors have published a similar story previously, however, given the importance of a better understanding of the underlying molecular mechanism for further research on both diseases, I believe this review has merit to be published.

However, I find a few critical remarks that need to be addressed.

In the Calcium toolkit section. The focus here is given to selected molecular players of calcium signaling. Especially in the figure, the information should be extended to provide functional correlates of calcium signaling (contraction, secretion, depolarization etc). This is important, as it allows the reader to correlate molecular changes with functional cellular alterations as found in WS and AD. The same is true for stimuli that induce calcium oscillations- are AD and WS cells insensitive to some of such stimuli? I would speculate that depolarization of cell membrane has a different effect on intracellular calcium dynamics in AD and WS (calcium is the only second messenger that can translate cells electrical activity into chemical signal). For this reason, it would increase readability if relevant cell stimuli that induce calcium oscillations are also added to Figure 1. As of now, this figure merely shows the subcellular localization of a few calcium-relevant proteins.

in the section 1.2. about mitochondrial Ca++ handling. The section is focused on transport of calcium ions INTO mitochondria. But how do calcium ions get out of mitochondria? Especially in view of negative mitochondrial membrane potential.  This is not clear in the current version and should be explained. 

In the Alzheimer section. It is clear that the authors want to promote the calcium theory of AD.

Paragraph on CALHM1 channel and its relation to AD (lines 239-247) it is stated that AD causing mutations in this gene lead to a reduction in cytosolic calcium levels. Quite opposite to what is described about cytosolic calcium in AD and WS later within the manuscript. However, no attention is made to this discrepancy. The question arises- is it elevation in cytosolic calcium levels that is a hallmark of AD and WS pathology, or is it plasma membrane (voltage dependent) Ca conductance? Please elaborate on this contradictory issue. 

In section 2.2. about ER role in AD. I wish authors to add summarizing statement, is it alterations of calcium signaling that lead to aberrant processing of amyloid peptides, or is it toxic beta amyloid that induces alterations of calcium handling? Or is it synergistic feed-forward vicious loop of these two processes? Right now the section is mixed with examples of both mechanisms, which contradicts the calcium hypothesis of AD but is presented for its support.

In section 3, about Wolfram syndrome. The molecular function of CISD2 protein is very well established, however it is not mentioned here- it binds iron-sulphur clusters (ISC) and takes part in their transport out of mitchondria. What is described in this manuscript is in effect calcium related consequences of disturbed  ISC transport. The function of Wfs1 is still unknown.

For the calcium role in WS (section 3.2.), the role of Wfs1 as revealed by oocyte electrophysiology was to increase calcium conductance (reference 141, lines 449-453). Interestingly, experiment of such kind was never repeated in subsequent studies.  According to this lonely report, Wfs1 forms a calcium channel and increases calcium conductance. WS is autosomal recessive disorder, with many identified mutations that include also frame shifts and loss of most of the protein.  Nevertheless, an increase in calcium permeability of ER membrane is reported as a consequence of Wfs1 deficiency, leading to an increase in cytosolic calcium. Therefore, ryanodine receptor antagonist is proposed as treatment of WS to ameliorate "leaky ER membrane". If Wfs1 forms a Ca channel or increases IP3R conductivity, how Wfs1 absence can lead to increased calcium permeability of ER membrane? Please explain.

Dandrolene is mentioned on line 472, this FDA approved drug has been applied to many WS patients in clinical trial at Washington University by Urano group, this clinical trial started about five years ago. A sentence regarding efficiency of RYR inhibition in WS patients must be included. This would provide a direct correlate of in-vitro data to real life situation. How beneficial effect of RYR inhibition in-vitro (line 472-473) correlates with beneficial effects of same drug in WS patients? This would also be strong support for calcium hypothesis of  WS and by analogy AD. Will dantrolene be effective against AD?

In the summary, line 554-557. It is nicely described that sigma receptor is related to every disorder and is therpeutic target for every disease. Its activation can be protective for AD, consequently sigma receptor agonist prevents beta-amyloid toxicity in WT neurons.  To my view, there is a logical conundrum here- applicability and validity of calcium theory is tested and proved by beta amyloid theory of AD.  Please elaborate on that- what is egg and what is henn (see also above). The summary is vague and lacks further directions- where shall we look for drugs for AD and WS (besides sigmaR)? Will drugs for AD be effective for WS? Is 3xTg mice a model of WS? or is WS mice good model of AD? Do WS mice or postmortem WS brains display formations of amyloyd plaques? They should, if amyloid formation is a consequence of abberant calcium signaling. 

Most importantly, as this is review that aims to put calcium signaling at the center of neurodegeneration (at least AD and WS). There is a number of calcium channelopaties, including mutations in RYR. Please include a short section summarizing the findings from these different calcium channelopaties. For example, mutations in RYR are bound to result in appearance of AD or WS symptoms in addition to arrhytmias, if calcium hyothesis of AD is valid. This will provide further support for unifying role of calcium in neurodegeneration and may reveal directions for further research.

And lastly, a summary drawing where unifying role of calcium in cytopathology of AD and WS is demonstarted is highly desirable. That could include possible treatment targets. 

Sincerely,

reviewer

Round 2

Reviewer 2 Report

All my previous questions are answered to the full extent. I recommend publishing this manuscript in this form.